# Hydrogen Sulfide Enhances Plant Tolerance to Waterlogging Stress

**DOI:** 10.3390/plants10091928

**Published:** 2021-09-16

**Authors:** Yaoqi Li, Da Sun, Ke Xu, Libo Jin, Renyi Peng

**Affiliations:** Biomedicine Collaborative Innovation Center of Wenzhou, Engineering Laboratory of Zhejiang Province for Pharmaceutical Development of Growth Factors, Institute of Life Sciences, Wenzhou University, Wenzhou 325035, China; 20461337004@stu.wzu.edu.cn (Y.L.); 20170033@wzu.edu.cn (D.S.)

**Keywords:** hydrogen sulfide, waterlogging, reactive oxygen species, gas signaling molecule, hypoxia tolerance

## Abstract

Hydrogen sulfide (H_2_S) is considered the third gas signal molecule in recent years. A large number of studies have shown that H_2_S not only played an important role in animals but also participated in the regulation of plant growth and development and responses to various environmental stresses. Waterlogging, as a kind of abiotic stress, poses a serious threat to land-based waterlogging-sensitive plants, and which H_2_S plays an indispensable role in response to. In this review, we summarized that H_2_S improves resistance to waterlogging stress by affecting lateral root development, photosynthetic efficiency, and cell fates. Here, we reviewed the roles of H_2_S in plant resistance to waterlogging stress, focusing on the mechanism of its promotion to gained hypoxia tolerance. Finally, we raised relevant issues that needed to be addressed.

## 1. Introduction

Hydrogen sulfide (H_2_S) is a colorless, toxic gas readily soluble in water with a pungent odor of rotten eggs [1,2]. It can be ionized to H^+^, HS^−^, and S^2−^ in aqueous solution; although the HS^−^ cannot cross the cell membrane, H_2_S, acting as a small liposoluble molecule, is five times more soluble in lipophilic solvents than in water, can permeate the lipid membrane freely, because of H_2_S’s poor water solubility, it is difficult to long-distance transport; however, SO_4_^2−^ and sulfur compounds can be realized long-distance transport through xylem vessels to participate in endogenous H_2_S metabolism in plant cells” [3,4,5]. H_2_S is widely regarded as a harmful gas produced in industrial production until the middle of the late 1990s. The physiological role and importance of H_2_S was gradually recognized by researchers. It was discovered that H_2_S can be produced by mammals through cysteine metabolism. With the further study of H_2_S as a gas signal molecule, many new physiological functions of H_2_S- and H_2_S-induced effects are of increasing interest, and it is considered as the third gas signaling molecule, which can be produced internally and plays multiple physiological functions in plants and animals. In recent years, more and more excellent research works on H_2_S have been done, and it is clearer and clearer to the mechanism of crosstalk between H_2_S with other molecules to regulate plant growth and development [2,6,7].

Water condition, affecting plant morphology, physiological, biochemical metabolism, and geographical distribution, is one of the important environmental factors for plant growth [8,9,10,11]. As the global warming, distribution of rainfall is seriously uneven, which leads to frequent waterlogging, and plant’s growth and development are impacted, especially xerophytes [9,12]. During the waterlogging period, although most vascular plants were obviously damaged and even died, the main cause of waterlogging damage to plants is not the water itself but secondary stress induced by excessive water [13,14]. Researchers initially found that plants accumulate higher than normal levels of H_2_S when exposed to stress. With the continuous in-depth study of the mechanism of stress response by many researchers, more and more evidence show that H_2_S plays a very important and extensive role in many stress processes in plants, including hypoxic stress induced by waterlogging [15,16].

A large number of studies have confirmed that H_2_S participates in the regulation of plants response to drought [17,18,19], extreme temperature [20,21,22,23], high salt [24,25], ultraviolet light [26], high osmotic pressure [27,28,29] and heavy metal (cadmium, chromium, lead, copper, mercury, etc.) [30,31,32,33,34,35,36,37,38] and significantly improves the stress tolerance of plants. Because endogenous H_2_S can be produced in plant cells, and H_2_S’s functions could be strengthened with the application of exogenous H_2_S. In order to study expediently, the application of H_2_S donors to plants has become a common method, especially when H_2_S responds to abiotic stresses. Waterlogging stress, which H_2_S is involved in response to in plants, acts as abiotic stress with significant destructive power, and this review will focus on the role of H_2_S as a signaling molecule in plant response to hypoxia stress induced by waterlogging [3,39,40].

## 2. The Role of H_2_S in Response to Hypoxia Stress Induced by Waterlogging in Plants

### 2.1. Multiple Factors Affect Tolerance to Waterlogging Stress

Waterlogging stress leads to the decrease in oxygen concentration in the rhizosphere of plants, resulting in the formation of hypoxia and anoxia. It directly acts on roots, making root hypoxia, nutrient absorption efficiency decreases, reduction in growth rate, further influences growth, and development of plants [41,42,43,44]. What is more, after a long period of evolution, plants have developed a series of stress response mechanisms. After sensing the anaerobic signal, a series of morphological, anatomical, physiological, and metabolic changes are caused by regulating the expression of genes so as to improve the ability to tolerate hypoxia and maintain the survival of individuals [45,46,47,48,49].

Plant waterlogging resistance is controlled by a combination of multiple factors [50,51,52,53,54]. The specific adaptive mechanism involves secondary signal transduction, gene expression, and protein synthesis, antioxidant enzyme system, and fermentation pathway under hypoxia stress [55,56,57,58]. Plants adapt to hypoxic stress from two aspects: avoidance and tolerance, that is, with promoting oxygen absorption and transport in vivo and reducing oxygen loss to escape hypoxic state, and with adjusting biochemical mechanisms to reduce the damage caused by hypoxia [47,59,60] when H_2_S acts a remarkable biological function in both these avoidance and adaptation strategies [6,61].

### 2.2. Synthesis of H_2_S in Plants

Because of toxicity for the over-accumulation of H_2_S in plant cells, different enzymes could be able to regulate the balance of H_2_S content [3,7,61,62]. In mammals, H_2_S production occurs in the cytoplasm and depends on the catalysis of enzymes, including cythione -β-synthase (CBS) and cythione-γ-lyase (CSE), in the sulfur-conversion pathway. On the one hand, CBS catalyzes the β-cysteine or homocysteine to produce cysteine and H_2_S. On the other hand, CSE, acting as a homologous tetramer enzyme, directly binds and catalyzes homocysteine and cysteine to product H_2_S. Additionally, 3-mercaptopyruvate thitransferase also contributes to the production of endogenous H_2_S from 3-mercaptopyruvate [63]. However, the key plant H_2_S synthases, named L/D-cysteine desulfhydrases (L/D-CDes), was first discovered in tobacco and gourd cells and mainly distributed in chloroplasts, mitochondria, and cytoplasm. Then, D-cysteine desulfhydrase (D-CDes) were identified and purified for the first time in Arabidopsis, which was found mainly based on homology characteristics similar to the D-CDes active protein in the large intestine. D-CDes are mainly located in the mitochondria, and their mRNA transcription level gradually increases plant growth and development but decreases during plant aging process. They also can catalyze the degradation of D-cysteine to produce H_2_S, pyruvate, and ammonia (Figure 1), and synthesis of H_2_S can achieve self-regulation when plants face different needs such as growth, fruit ripening, diseases, and insect pests and abiotic stresses, and acquire regular growth and development or stress tolerance [64,65,66,67].

### 2.3. Intermediate Metabolite of H_2_S in Plants Promotes Stress Tolerance

The sulfur-containing defense system of plants includes elemental sulfur, H_2_S, glutathione (GSH), plant chelating agents, various secondary metabolites, and sulfur-rich proteins [68,69]. SO_4_^2−^ is absorbed and enters the vacuole to regulate the osmotic pressure of the cells, and some are transported to the aboveground part of the plant by high-affinity transporters and enters into the chloroplasts, chromoplasts, and other plastids to participate in the anabolism of SO_3_^2−^, these also could occur in the cytoplasm [70,71,72,73]. The formation of ASE and Cys from carrier compounds of H_2_S or sulfur with OAS under OAS-TL can occur in both cytoplasm and mitochondria. The Cys is the metabolic precursor of glutathione, phytochemicals, and some sulfur-rich proteins, which is of great significance for the normal physiological activities of plants (Figure 2) [32,74].

Cysteine is also the metabolic precursor of many important molecular substances such as vitamins, cofactors, antioxidants, and many defense substances, and it could be further metabolized into other sulfur-rich proteins (SRPs), plant chelating peptides and GSH, and so on [72,75]. Meanwhile, O-acetylserine(thiol)lyase isoform a1 (OAS-A1), the main isozyme of OAS-TL, and L-cysteine desulfhydrase 1 (DES1), a cysteine-degrading cytoplasmic thiol enzyme, affected the homeostasis of cysteine in the cytoplasm. In this series of metabolites, H_2_S is a very important intermediate product in the sulfur metabolism pathway, and sulfur metabolism has a significant impact on the stress adaptability of plants [76].

## 3. Adventitious Root Formation, Photosynthesis Efficiency Improvement, Cell Death Alleviation Promoted by H_2_S against Waterlogging Stress

### 3.1. H_2_S Enhances the Occurrence of Adventitious Roots

The occurrence of adventitious roots is one of the ways for plants to adapt to low oxygen stress in waterlogging [77,78,79]. H_2_S produced by microorganisms in soil is little absorbed by roots for its poor water solubility, and the main pathway of H_2_S accumulation is endogenous production in root cells. H_2_S can promote the elongation of plant roots; for example, a low concentration of exogenous H_2_S (0–40 μmol · L^−1^) can promote the growth of pea radicles and the formation of adventitious root formation in cucumber [80,81]. It was also found that the endogenous H_2_S, indoleacetic acid (IAA), and NO contents in sweet potato stem tip increased in sequence with the addition of H_2_S donor sodium hydrosulfide (NaHS), suggesting that H_2_S may induce adventive root formation through IAA and NO [67,82,83].

### 3.2. H_2_S Elevates the Photosynthetic Efficiency of Plants

Photosynthetic efficiency is also an important reference index for the degree of hypoxia stress of plants subjected to waterlogging [84]. As early as 1973, Gassman reported that H_2_S could break the disulfide bonds of photosystem proteins in yellow bean leaves and make them reversible. Then, much evidence showed that exogenous H_2_S could increase the chlorophyll content of plant leaves [85,86]. For instance, H_2_S increases the chlorophyll content of spinach leaves, changes the chloroplast structure, and improves the photosynthetic rate, which may be through the regulation of rubisco activity and the redox modification of sulfhydryl compounds to enhance photosynthesis [84]. Photosynthesis could be promoted with H_2_S through promoting photosynthetic enzyme expression, chloroplast biogenesis, and thiol redox modification in *Spinacia oleracea* seedlings [19].

### 3.3. H_2_S Alleviates Plant Cell Death

Cell is the basic constituent unit of the plant, whose surviving or not directly determines the survival state of plants during waterlogging [87]. Further, though the factors influencing cell survival include cytoplasmic redox state, pH, energy supply, metabolic enzyme activity, programmed cell death factor, and many other factors, H_2_S is generally involved in these physiological and biochemical processes [3,88,89]. Strawberries smoked with H_2_S have been demonstrated that exogenous H_2_S could maintain low rot index, high fruit firmness, low respiration density, and polygalacturonase activity, thus extending the fresh-keeping period of strawberries after picking [90]. There are studies that found that H_2_S can reduce the plant tissue and cell death against hypoxia stress induced by waterlogging in pea, maize, and Arabidopsis, respectively [16,91,92].

## 4. How H_2_S Enhances the Hypoxia Tolerance of Plants during Waterlogging

### 4.1. H_2_S Enhances the Activity of Antioxidant System to Gain Waterlogging Tolerance

Studies have shown that abnormal levels of reactive oxygen species (ROS) will be produced in plants subjected to stresses, resulting in oxidative damage of cells and some related key enzyme genes up-expression observably. Catalase (CAT), peroxidase (POD), and superoxide dismutase (SOD) are core members of the antioxidant enzyme system and have been recognized as key players in the complex signaling network in plants’ response to environmental stresses [93,94,95]. Previous studies showed that the activities of CAT, POD, and SOD were enhanced under mild waterlogging conditions, while their activities increased first and then decreased under severe waterlogging conditions [14]. Under waterlogging stress, the activities of POD and SOD in leaves of begonia seedlings increased significantly at the initial stage and then tended to be similar to the control [96]. The activities of key antioxidant enzymes in root tip cells of pea and maize treated with H_2_S were significantly higher than untreated groups, and the degree of cell death was less than untreated groups, which was also reappeared in seedlings of *Arabidopsis thaliana* [91,92,97].

Reduced glutathione/oxidized glutathione (GSH/GSSG) is a meaningful parameter to reflect the redox state of cells (Figure 3) [98,99]. NaHS pretreatment could recover the loss of ascorbic acid (AsA) content and further increase the GSH content under extreme environmental conditions [100,101]. Furthermore, the content of GSH was further increased to maintain the ratio of AsA/dehydroascorbate (DHA) and GSH/GSSG, balance the content of mineral elements, reduce the absorption of Na^+^ and the ratio of Na^+^ /K^+^, and increase the endogenous H_2_S content to protect chlorophyll, carotenoid and soluble proteins from oxidative damage [102,103]. It was found that NaHS up-regulates the activity of glyoxalase I (Gly I) and glyoxalase II (Gly II) enzymes related to glycine (Gly) metabolism in rice, thus maintaining the GSH system homeostasis and slowing down the cytotoxicity of methylglyoxal (MG) and ROS [17,104]. Oxidative damage caused by water stress could be alleviated by regulating ascorbic acid and glutathione metabolism resulting in alleviation of the impact of water stress on wheat seedlings [68,105]. It is worth noting that exogenous NaHS (H_2_S donor) was applied to maize seedling roots inducing improvement of endogenous NO level in root tip cells, and NO acted as a second messenger to enhance the cyclic metabolism of AsA-GSH, maintained antioxidant system capacity, reduced ROS-induced macromolecular damage, and alleviated the damage caused by peroxidation to plants [106,107].

### 4.2. The Crosstalk between H_2_S and Hormones Improves the Hypoxia Tolerance

H_2_S interaction with plant hormones withstands waterlogging-induced hypoxia stress. The physiological activities of higher plants are in a complex signal network, and there are different interactions between different pathways to jointly resist abiotic stress, and H_2_S changes the balance of different signal substances and regulates plant growth and stress tolerance [31,108]. For example, salicylic acid (SA) is a phenol signaling substance that acts upstream of H_2_S and participates in plants' response to stress [23,109]. Moreover, H_2_S may be involved in the stomatal closure process induced by abscisic acid (ABA), ethylene and jasmonic acid (JA), and exogenous ABA can significantly improve the H_2_S level and L/D-cysteine desulfhydrase activity in leaves, while H_2_S synthesis inhibitors can reverse the effects of ABA [17,110]. In the study of maize seedlings, H_2_S may act as a downstream signal molecule of NO to respond to waterlogging-induced hypoxia stress [92].

### 4.3. H_2_S Affects Respiratory Metabolism to Improve Hypoxia Stress Tolerance

Respiration is the most basic source of power to maintain plant metabolism, growth, and transform nutrients. The plant root respiration is closely related to plant matter metabolism and energy metabolism, and the success of root respiration is one of the important indicators to measure plant root normal function and stress tolerance [40,111]. Malic dehydrogenase (MDH), phosphofructokinase (PFK), and glucose-6-phosphate dehydrogenase (G-6-PDH), whose activities directly affect the respiratory rate, are the key enzymes that regulate the rate of each respiratory metabolic pathway. Additionally, NaHS treatment of chestnut roots could improve the enzyme activities of MDH, PFK, and G-6-PDH to a certain extent, thus improving the stress tolerance of plants [84,112].

### 4.4. Sulfur-Sulfhydrylation of Proteins by H_2_S Strengthens Hypoxia Tolerance

In plant cells, H_2_S firstly modifies Cys residues with thiol, and then the Cys-SH group is transformed into the Cys-SSH sulfyl group, which directly regulates the activity of proteins [113]. H_2_S induces changes in actin cytoskeleton and inhibits actin polymerization through s-vulcanization of actin, proving that H_2_S regulates actin dynamics and affects root hair growth. Another study showed that exogenous H_2_S promoted the sulfhydrylation of ascorbate peroxidase (APX) in *Arabidopsis thaliana* and improved its activity [72,114]. In conclusion, S-sulfhydrylation is an integrant pathway for an H_2_S signaling molecule to exert biological activity in plants, and proteins, modified with sulfhydrylation, provide direct or indirect support for the acquisition of tolerance to low oxygen stress in waterlogging.

### 4.5. H_2_S Associating with Ca^2+^ Elevates Hypoxia Tolerance

The increase in Ca^2+^ concentration is similar to the accumulation of ROS, which is one of the basic steps in plants' response to stress signals. Just like other signaling molecules, Ca^2+^ is a necessary universal second messenger acting as transduction and regulatory factor in plants, which can produce adaptive responses by Ca^2+^ binding proteins that sense rapid Ca^2+^ increase and transmit specific signals [115,116]. The reduction effect of H_2_S is related to the promotion of Ca^2+^ influx, and H_2_S generated by CBS was 3.5 times when there was Ca^2+^/calmodulin (CaM) more than without them, while that can be inhibited with treating CaM inhibitors [117]. It has been reported that the application of exogenous Ca^2+^ and its ionic carrier A23187 could significantly enhance the antioxidant capacity induced by NaHS. However, Ca^2+^ chelating agents, ethylene glycol diethyl ether diamine tetraacetic acid (EGTA), plasma membrane channel blocker La^3+^, and calmodulin antagonist chlorpromazine and trifluoperazine can weaken this resistance [118].

### 4.6. H_2_S Involves in Regulating Gene Expression to Improve Tolerance to Waterlogging Stress

The hypoxia stress of waterlogging, as one stress of plants response to, is regulated by many types of genes and involved many kinds of stress regulation processes [12,45,46]. H_2_S up-regulated genes including SICDKA1, SICYCA2, CYCD3, CDKA1, ARF4, and ARF7 associated with cell cycle-related to lateral root growth in tomato seedlings, suggesting that H_2_S and indoleacetic acid (IAA) jointly induced lateral root formation in tomato seedlings, which can enhance both drought resistance and waterlogging tolerance [83,119]. Transcriptomic sequencing of *Arabidopsis thaliana* pretreated with H_2_S under hypoxia conditions found that significant changes related to transcription regulation-associated genes, hypoxia-sensing genes, and hormone signal transducer genes [16,54]. In addition, the accumulation of H_2_S helps maize seedlings to enhance the waterlogging tolerance, during when the expression of related genes contained stress response, hypoxic induction, energy metabolism, and other changes remarkably [92].

Based on the above descriptions, enhancement of plant waterlogging tolerance involved in the regulation of H_2_S was not determined by a single factor, but rather through gene expression, protein modification, regulation of plant hormones, and interaction with other signaling molecules, which were reflected in the reduction in cell death, enhancement of plant photosynthesis, formation of lateral roots at the macro level. In the future, through making full use of advanced gene editing and proteomics technology, the mechanism of H_2_S participating in the regulation of plants to improve the waterlogging tolerance will be continuously improved.

## 5. Conclusions and Perspectives

The function of H_2_S runs through the whole process of plant growth and development, and it is also indispensable in responding to hypoxic induced by waterlogging (Figure 4). In this review, we summarized the roles of H_2_S in adventitious root formation, photosynthesis efficiency improvement, and cell death decrease in plants responding to waterlogging hypoxia stress and further discussed the specific approaches from the perspective of molecular mechanisms.

In spite of these advances in the exploration of H_2_S's response mechanism to hypoxia, many challenges remain. Firstly, the direct target, upstream and downstream cascade reaction of H_2_S in the plant signal transduction process, and the crosstalk among H_2_S and other signal molecules should be further elucidated. On the other hand, the process of plant response to hypoxia stress depends on the proportion of different hormones, of which H_2_S may be the intermediate to coordinate the interaction, while whose mechanism is still not clear yet. However, with the continuous development of transgenic technology and gene-editing technology, we believe the molecular mechanism of H_2_S involved in the regulation of hypoxia response is becoming increasingly clear.

## Figures and Tables

**Figure 1 plants-10-01928-f001:**
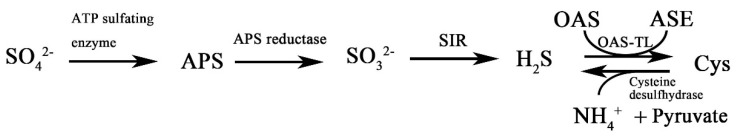
Synthesis of H_2_S in plant cells [15]. APS: 5’-adenylylsulfate; SIR: sulfite reductase. OAS: O-acetyl serine; OAS-TL: O-acetyl-L-serine (mercaptan) lyase; ASE: acetate; Cys: cysteine.

**Figure 2 plants-10-01928-f002:**
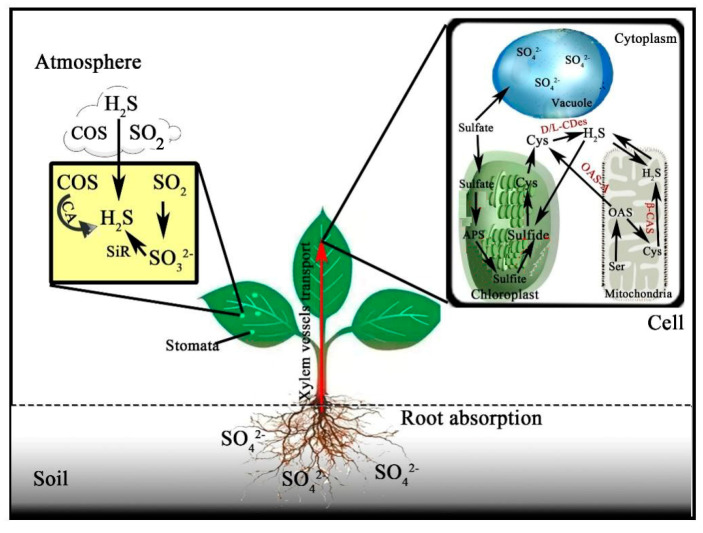
H_2_S plays a pivotal role in sulfur metabolism in plants. Sulfur elements needed for sulfur metabolism in plants are mainly absorbed by roots in the manner of SO_4_^2−^ from the soil, which was transported to various parts of plants through microtubules to participate in the process of sulfur metabolism in plants. Another complementary pathway is the absorption of H_2_S, carbonyl sulfide (COS), and SO_2_ from the air through stomata on the leaves, thus immobilization of sulfur into the sulfur metabolic pathway in plants. CA: carbonic anhydrase, SiR: sulfite reductase.

**Figure 3 plants-10-01928-f003:**
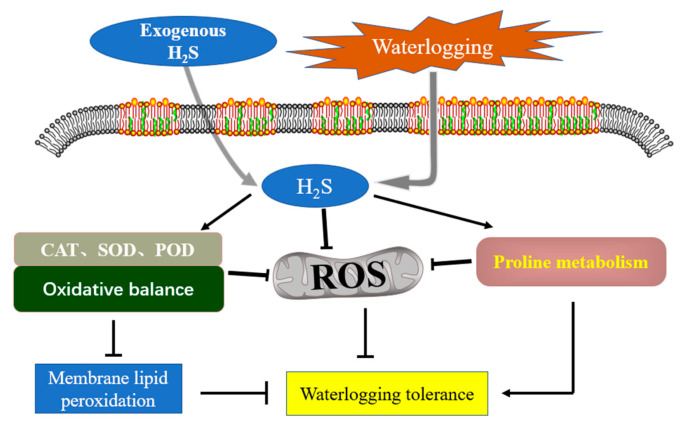
Schematic diagram of H_2_S enhancing the activity of antioxidant system to acquire the tolerance to waterlogging-induced hypoxia.

**Figure 4 plants-10-01928-f004:**
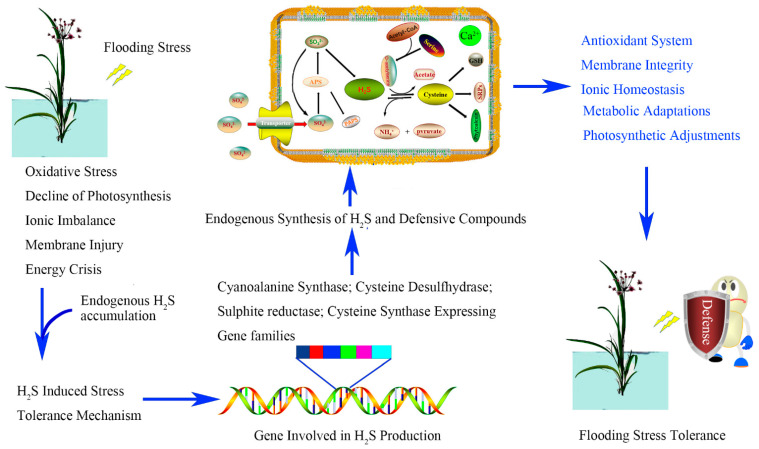
The role of H_2_S in the regulation of waterlogging stress in plants.

## Data Availability

Not applicable.

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
