# Peer review of "Hydrogen Sulfide Enhances Plant Tolerance to Waterlogging Stress"

_plants, 2021, doi:10.3390/plants10091928_

Round 1
Reviewer 1 Report
The manuscript is a review that systematically summarizes function of hydrogen sulfide in the enhancement of plant tolerance to waterlogging. In my opinion, the review is very well thought out, the diagrams are quite clear and well summarize the results of the analysis of literature data described in the text. This article made a positive impression on me.
Minor comments:
Line 20: There is ‘Waterlogging’ should be ‘waterlogging’
Line 57: There is ‘will lead’ should ‘leads’
Line 71: There is ‘hypoxic’ should be ‘hypoxia’
Line 81 There is figure 1 should be ‘Figure 1’
Line 90: There is ‘ ... some of it’ – is it correct?
Line 106: Figure 2: There is SO4 should be SO42-; Please explain CA and SiRPicture of chloroplast is not readable, especially red letters
Line 210 and 235: Please give Arabidopsis thaliana as italic
Line 256: Figure 4: The graph (containing molecules) isn’t readable
Reviewer 2 Report
I have a few suggestions:
1. "H2S is five times more soluble in lipophilic solvents than in water, which made it permeates the lipid membrane freely[3-5]." Solubility of H2S in hydrophobic solvents does not mean that this molecule can "freely permeate" biological membranes This should be clearly explained. Maybe You should mention something about the stability of this molecule in plant or animal cells as well, and how far this "signaling molecule" can be translocated in a plant?
2. line 100: "vitamins, cofactors, antioxidants and many defense substances, which can be further generated in other sulfur-rich proteins (SRPs)" This is something I do not understand...
3. line 174 : "H2S was applied to improve endogenous NO level," This statement is not clear.
Reviewer 3 Report
This seems to be a timely and important contribution complementing recent review papers on this fascinating gaseous signalling molecule. It seems that more praise should be given to previous great reviews on the subject, when introducing to historical development of studies with hydrogen sulfide.
I have two general comments on this review. First, I would like to see a clear distinction made between mammalian and plant functions, when describing role of several common signalling molecules. Second, I would like to see a clear distinction between effects of endogenous and exogenous hydrogen sulfide throughout the paper. Many cited sources, for example, on "participation" of H2S in the regulation of plant responses report results from experiments where exogenous H2S (through specific donors) was used. I suggest that it needs to be made clear initially that plants indeed produce this compound, but experimental approaches usually involve exogenously increased H2S concentration, which could lead to totally or partially different effects due to differences in compartmentation and localization of respective receptors. May be it is a good idea to cover methodological aspects of studies involving H2S at the very beginning of the review (for example, at the end of introduction?). In this respect, scheme depicting the role of H2S in regulation of flooding stress responses (not "flooding stress" as it is stated!) needs to be made more comprehensible, as it is not clear from where "Exogenous H2S" appears. Is it produced by waterlogged soil through some microbiological processes? Moreover, there needs to be an explanation somewhere in the text on the possible autoregulative effect of H2S on its own biosynthesis. At present, picture in Fig. 4 tells us that it is exogenous H2S on the basis of flooding-stress altered metabolism, which induces expression of genes involved in endogenous H2S synthesis, further leading to its increased concentration and synthesis of defense compunds.
Attention should be paid to certain additional moments to obtain better clarity of the whole story. For the role of CO in plants, dedicated review need to be cited (Xuan et al. 2008; He, He 2014; Wang, Liao 2016). I would suggest to refrain from overgeneralization when speaking about role of CO in plants, as it is just emerging (not as "plays multiple physiological functions in plants").
Care needs to be taken to avoid possible misunderstanding when using term "water stress" along with "waterlogging" (Line 171), as the cited source [67] indeed describes effects of water shortage (water stress), while the source [104] deals with anoxia tolerance, a major factor during waterlogging.
From a substantive point of view, I could not find any information regarding plant morphological adaptations to waterlogging, as aerenchima formation (involving also programmed cell death) and leaf petiole elongation (involving ABA, ROS etc.). I just wonder if there is any evidence on involvement of H2S? Also, what about effects on induced glycolysis as related to temporar relief from oxygen shortage in roots (complementing subsection 4.3)?
Some parts of text need careful content-specific editing, as this sentence (Lines 173–177):
"It's worth noting that exogenous H2S was applied to improve endogenous NO level, and NO acted AsA second messenger to enhance the circulation metabolism of AsA-GSH, balance the REDOX state, reduce ROS mediated macromolecular damage, and alleviate the damage caused by peroxidation to plants[105, 106]". Please pay attention to words "it's" (it is?), "improve" (increase?), "AsA" (first mention, as a?), "REDOX" (redox?). By the way, what is meant by "balancing the redox state?"
Round 2
Reviewer 3 Report
Thank you for revisons performed. In general, i am satisfied with responses, but one particular point still needs to be clarified, namely, position of "Exogenous H2S" in Fig. 4. There is no particular response to my previous comment:
"... scheme depicting the role of H2S in regulation of flooding stress responses (not "flooding stress" as it is stated!) needs to be made more comprehensible, as it is not clear from where "Exogenous H2S" appears. Is it produced by waterlogged soil through some microbiological processes? Moreover, there needs to be an explanation somewhere in the text on the possible autoregulative effect of H2S on its own biosynthesis. At present, picture in Fig. 4 tells us that it is exogenous H2S on the basis of flooding-stress altered metabolism, which induces expression of genes involved in endogenous H2S synthesis, further leading to its increased concentration and synthesis of defense compounds."
If it is intended to incorporate results from studies without flooding but with H2S treatment, "Exogenous H2S" needs to be inserted in the scheme after "Endogenous synthesis" before reception and signalling cascade.
In addition, very careful language and style editing is necessary, as still there are numerous mistakes and stylistic problems throughout the manuscript.
